# Seismogenic Disturbances of the Ionosphere During High Geomagnetic Activity

**Aleksandr Namgaladze [1],*, Mikhail Karpov [2,3] and Maria Knyazeva [1]**

[1] Laboratory of Computer modeling of physical processes in the near-Earth environment, Murmansk Arctic State University, Murmansk 183038, Russia

[2] Kaliningrad Department of the Pushkov Institute of Terrestrial Magnetism, Ionosphere, and Radiowave Propagation (IZMIRAN), Russian Academy of Sciences, Kaliningrad 236016, Russia

[3] Laboratory of applied radiophysical research of the atmosphere and ionosphere, Immanuel Kant Baltic Federal University, Kaliningrad 236016, Russia

* Correspondence: namgaladze@yandex.ru

**Abstract:** Herein, we analyze the variations in the ionosphere for the period of two weeks before the M6.7 earthquake in India on 3 January 2016. The earthquake occurred after a series of magnetic substorms on 31 December 2015 and January 1, 2016. The relative total electron content (TEC) disturbances have been estimated using global TEC maps and calculated numerically using the 3D global first-principle Upper Atmosphere Model (UAM) for the whole period including the days before, during, and after the substorms. Numerical simulations were repeated with the seismogenic vertical electric currents switched on at the earthquake epicenter. The UAM calculations have reproduced the general behavior of the ionosphere after the main phase of the geomagnetic storm on January 1, 2016 in the form of negative TEC disturbances propagating from high latitudes, being especially strong in the Southern (summer condition) Hemisphere. It was shown that the local ionospheric effects of seismic origin can be identified in the background of the global geomagnetic disturbances. The seismo-ionospheric effects are visible in the nighttime regions with the additional negative TEC disturbances extending from the eastern side of the epicenter meridian to the western side, both in the observations and in the UAM simulations. It was found that the vertical electric field and corresponding westward component of the electromagnetic [E × B] drift played a decisive role in the formation of the ionospheric precursors of this earthquake.

**Keywords:** earthquake; total electron content; ionosphere; modeling; electric field

## 1. Introduction

In the paper by Namgaladze et al. (2013) [1], the total electron content (TEC) variations before strong earthquakes (EQs), modeled with the global upper atmosphere model (UAM), were considered for geomagnetically quiet conditions. It is standard practice to avoid mixing the geomagnetic and EQ effects. Of course, such mixing makes the problem of the EQ's influence on the ionosphere more difficult, but we hope that the differences in the time and spatial scales of the geomagnetic and EQ preparation phenomena can help us to resolve this problem.

The main features of the EQ preparation effects in TECs are the locality, connections with the tectonic faults, weak mobility, and small occupied areas [1] in comparison with the ionospheric effects of the geomagnetic storms and substorms. The latter are usually global, connected with the auroral precipitations and electric fields related to the solar wind, and propagate equatorwards from the auroral zones, sometimes in the form of Travelling Ionosphere Disturbances (TIDs) due to the Acoustic

Gravity Waves (AGW) (Brunelli and Namdaladze, 1988) [2]. Their duration can range from several hours for substorms to several days for storms.

The common features of the EQ and geomagnetic effects in TEC behavior is the geomagnetic conjugation along the magnetic field lines and their sensitivities to the electric fields and thermospheric $O/N_2$ composition variations.

In this study, we analyze the ionosphere behavior for the period of high geomagnetic activity on 31 December 2015 and January 1, 2016 followed by the M6.7 earthquake in India on 3 January 2016. The Upper Atmosphere Model (UAM), described in [1], is used for modeling in our analysis.

## 2. Experiments

The geomagnetic activity for the period from December 20, 2015 to 6 January 2016 is characterized by two series of disturbances on December 20–21 and December 31–January 1, and a third on January 6, i.e., three days after the M6.7 earthquake in India on 3 January 2016. The Kp magnetic activity index, describing the planetary disturbance of the Earth's magnetic field, was about 3–7 on December 20–21, 2015, 3–6 on 31 December 2015–January 1, 2016, and 2–4 on 6 January 2016. The AE index, describing the auroral magnetic activity, was from 200 to 1300 on December 20–21, 2015, from 200 to 1200 on 31 December 2015–January 1, 2016, and 150–500 on 6 January 2016. From December 25–30, 2015, geomagnetic activity was low.

According to the U.S. Geological Survey, this earthquake happened at 23:05:22 UTC on 3 January 2016 with the epicenter 30 km west of Imphal, India (24.8° N, 93.7° E). The epicenter location in the magnetic coordinates (15° mag.lat., 165° mag.long.) corresponds to the north crest of the well-known Equatorial Ionosphere Anomaly (EIA) [2], where the influence of EQs on the ionosphere is significant [1].

The global ionosphere maps of the total electron content (GIM-TEC) in the IONEX format obtained from the global navigation system satellites data [3] were used for analysis of the ionosphere variations for the whole period of two weeks preceding the earthquake. The GIM-TEC covers the ±87.5° latitude and ±180° longitude range with a spatial resolution of 2.5° and 5°, respectively, and 2 h time resolution. We estimated the relative TEC disturbances as

$$dTEC = (TEC - TEC_0)/TEC_0 \times 100\%, \tag{1}$$

where TEC denotes the value for the current moment and $TEC_0$ is the background value.

The choice of the background $TEC_0$ values is based on averaging the TEC for the nearest quietest period preceding the examined event—in our case, December 25–30, 2015.

For numerical calculations of the TEC variations, we use the three-dimensional global first-principle Upper Atmosphere Model (UAM) [1,2]. Aside from the continuity, momentum, and heat balance equations for neutral and charged gas components, the UAM also calculates the electrostatic potential pattern by numerically solving the electric potential taking into account electrostatic and dynamo electric fields, the magnetospheric, and seismogenic electric currents:

$$\nabla[\sigma^T(\nabla\varphi - [V \times B]) - j_m - j_s] = 0, \tag{2}$$

where $\sigma^T$ is the ionosphere conductivity tensor, $\varphi$ is the potential of the electrostatic field, V is the velocity vector of the neutral gas motion, B is the magnetic induction vector, and $j_m$ and $j_s$ are the density of magnetosphere and seismogenic electric currents, respectively.

The seismogenic $j_s$ and magnetospheric $j_m$ electric currents are added at the lower and upper boundaries of the electric potential equation, i.e., at 90 and 175 km, respectively.

Magnetospheric currents are the field-aligned currents of zones 1 and 2 (FAC1 and FAC2, respectively) flowing between the ionosphere and the magnetosphere. It is assumed that at 175 km, the areas with the FACs, flowing into the ionosphere and out of it, coincide with the position of the auroral oval boundaries on the morning and evening sides (06:00 and 18:00 MLT, respectively).

The position of the oval is set up depending on the Kp index [4]. The FAC1 density values have been selected iteratively, until the electric potential difference across the polar cap did not reach the value calculated depending on the AE index [5]. The FAC2 density values are assumed to be equal to 0.7·FAC1.

The seismogenic electric currents $j_s$ were added to the potential equation locally, above the Indian earthquake epicenter. Previously, $j_s = 20$ nA/m$^2$ has been used as the density of the vertical electric currents in the UAM numerical calculations for the middle-latitude earthquakes which occurred during quiet geomagnetic activity, and the TEC disturbances were ~40% relative to the non-disturbed values [6,7]. The chain of electric current sources with the length of 4000 km was setup along the tectonic fault, parallel to 30° meridian.

In the present calculations, two current density values were used: $j_s = 20$ nA/m$^2$ and $j_s = 40$ nA/m$^2$. The last value is 1000 times higher than the fair-weather currents and about the same order of magnitude as the thunderstorm currents densities [7]. In the calculations of present model for the period from 25 December 2015 to 3 January 2016, including the disturbances of December 31–January 1, we used the spatial distribution of the FACs dependencies on AE and Kp as described above and the spatial distributions of the precipitating electron fluxes depending on Kp according to the statistical model of auroral electron precipitation [4].

Calculations were performed in two variants: The self-consistent version numerically solved all theoretical equations; and the partly empirical version used data for the neutral gas from the NLRMSISE-00 empirical model of the thermosphere [8]. In the second version, Ap index was set up as an additional input parameter for modeling the thermosphere behavior during the geomagnetic disturbances of December 31–January 1.

## 3. Results

The GPS observed and UAM calculated TEC disturbances relative to the background values at 10:00–14:00 UT on January 1, 2016 are presented in Figures 1 and 2. This day was chosen for modeling due to the fact that it is there that we see the typical local EQ precursor spots in TEC only on January 1, 2016. The model results were obtained using both versions of the UAM: the self-consistent version (hereafter, UAM-T) and UAM with the NRLMSISE-00 (hereafter, UAM-M). In both cases, the numerical calculations were carried out taking into account seismogenic currents as well as discounting them; this was to distinguish the effects of the seismogenic electric currents from the effects of magnetic activity, but this is not related with the earthquake preparation.

The maximum point of the geomagnetic storm main phase on January 1, 2016 is at 00:00 UT. The ionosphere effect of the geomagnetic activity pronounces itself most clearly after the main phase, both in the GPS observations and UAM calculation results (both in UAM-T and UAM-M; however, in UAM-T this ionosphere storm effect is much larger than in UAM-M). This effect has the form of the negative ionospheric phase (the TEC decrease relative to the background, quiet values) in the Southern (summer) Hemisphere due to the thermosphere motion from the high latitudes toward the equator and downward motion of plasma from the plasmasphere to the ionosphere. The resulting effect is the decrease in the concentration ratio between the atomic and molecular components of the neutral gas ($O/N_2$ ratio) which leads to the increase in the ion recombination rates, and eventually, to the decrease in the electron density and TEC. The negative phase propagates from the high latitudes to, at least, the epicenter latitude, and TEC disturbances reaches –50% according to the observations (Figure 1a).

The addition of the seismogenic electric currents flowing upward (charging the ionosphere) drastically changes the calculated TEC pattern during the nighttime in contrast to the daytime due to the higher electric conductivity of the daytime ionosphere. The additional negative TEC disturbances appear in the area between the latitudes of the epicenter and the magnetically conjugated point, ±30° to the east and to the west of the epicenter meridian. The GIM-TEC also shows a similar pattern in the same area (see Figure 1a). The comparison of the results obtained with the seismogenic electric current density $j_s = 20$ nA/m$^2$ and $j_s = 40$ nA/m$^2$ shows that the seismogenic TEC disturbances stay

pretty much the same in both calculation variants, except the magnitude of disturbances increases with the doubling of the current density.

Thus, the seismo-ionospheric effects of the Indian earthquake preparation manifest themselves both in the observations and calculations simultaneously with the effects of the geomagnetic substorm, and they are comparable with each other by the order of the magnitude. The UAM-T calculations presented in Figure 1 agree with the observations better than the UAM-M calculations in Figure 2. GPS observations also show the strong positive TEC disturbances exceeding 80% and located northeast of the epicenter. They are similar to the seismogenic disturbances reported in [6,7], with the exception of the magnetic conjugation and daytime appearance. An increase in the vertical electric currents does not lead to the appearance of the strong positive TEC disturbances in that area in either the UAM-T or the UAM-M calculations.

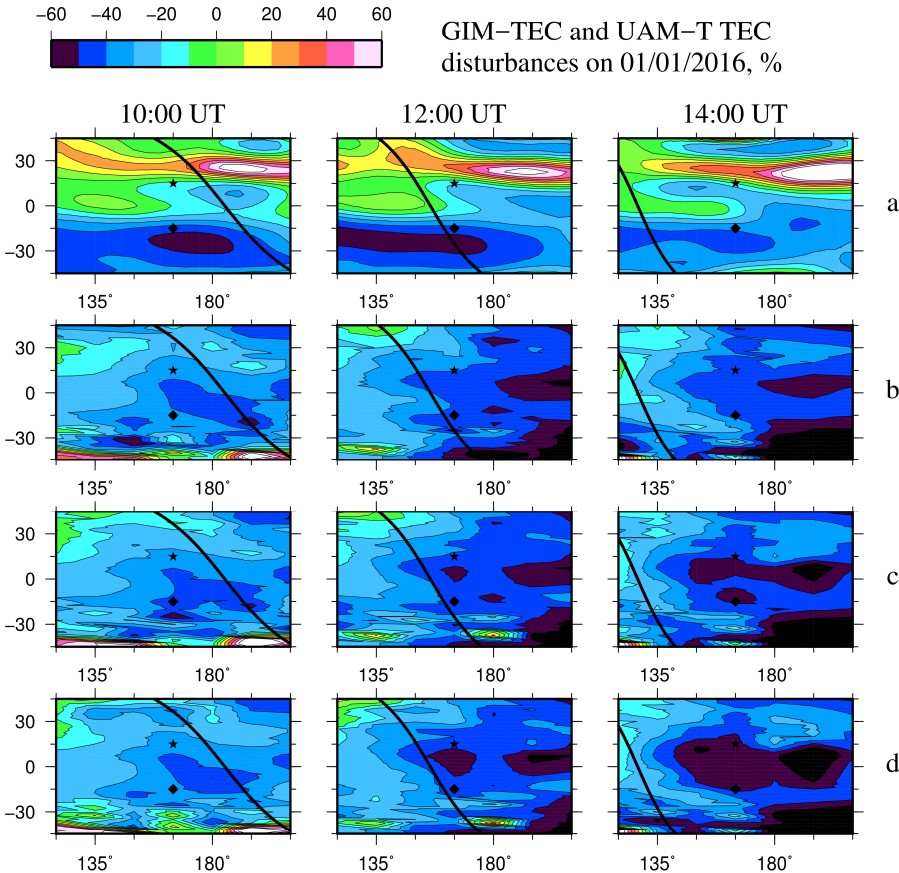

**Figure 1.** GPS observed and UAM-T (self-consistent UAM version) calculated total electron content (TEC) disturbances (%) relative to the background values in the geomagnetic coordinates at 10:00–14:00 UTC on January 1, 2016: (**a**) Global ionosphere maps of the total electron content (GIM-TEC); (**b**) UAM-T calculated without currents; (**c**) UAM-T calculated with $j_s = 20$ nA/m$^2$; (**d**) UAM-T calculated with $j_s = 40$ nA/m$^2$. Black line denotes the terminator, black star and diamond represent the earthquake epicenter and magnetically conjugated point, respectively.

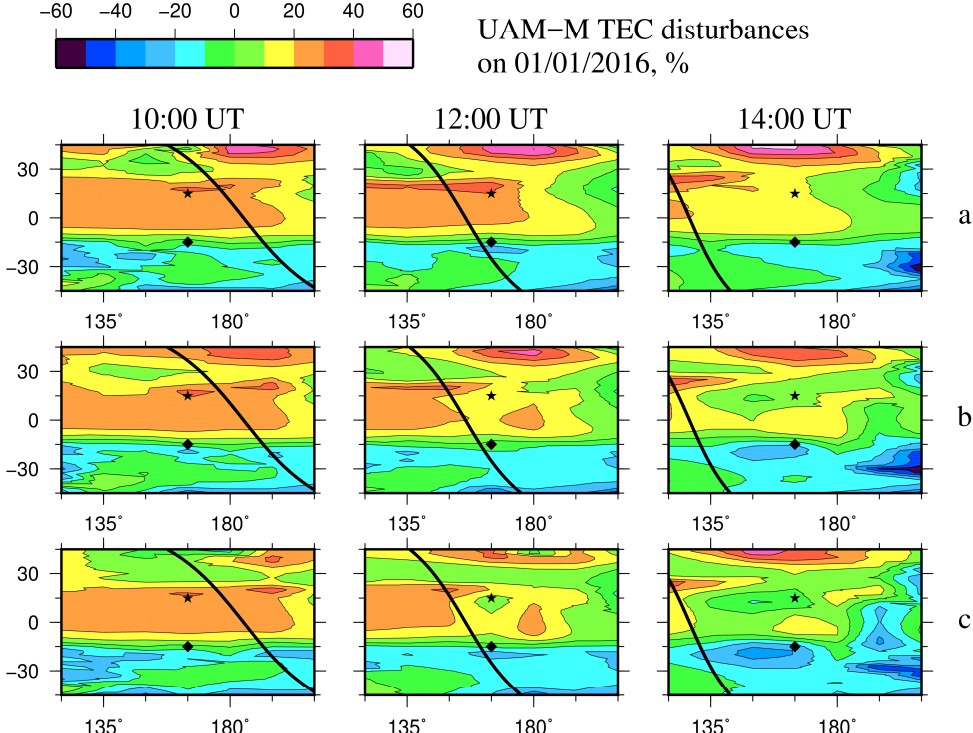

**Figure 2.** The same as Figure 1, but for the UAM-M (UAM version with the empirical model NRLMSISE-00 for neutral gas) calculations: (**a**) Without currents; (**b**) $j_s = 20$ nA/m$^2$; (**c**) $j_s = 40$ nA/m$^2$.

## 4. Discussion

The presented results show that there is a clear difference from the results obtained previously using the same method but for other conditions: 1) The low latitude EQ location instead of the middle latitude location; and 2) the high magnetic activity instead of the low magnetic activity.

In the previous UAM calculations [6,7,9–11], where the middle-latitude earthquakes were simulated, the main cause of the TEC disturbances was attributed to the electrostatic electric field generated as a result of the seismogenic vertical electric current. The electrostatic field was directed radially from the epicenter in the case where the electric currents flow to the ionosphere, and toward the epicenter for the currents of the opposite direction. Thus, the zonal component of the electrostatic field flowed in opposite directions on opposite sides of the epicenter meridian. The eastward component led to the upward plasma drift, and correspondingly, to the positive TEC disturbances; the westward electric field caused the downward plasma motion, and thus, the negative TEC disturbances.

In the present case study, we consider the low-latitude earthquake, located near the north crest of the EIA; and herein, we deal mainly with the dynamo electric field of induction origin, dominating at low latitudes in comparison to the middle latitudes [2]. This is added to the electrostatic field, and thus, they both create the new electric potential and corresponding drift velocity patterns.

The zonal drift velocity patterns at 300 km are presented in Figure 3 for the UAM-T calculations. The differences between the UAM-T and UAM-M modeling results for the zonal drift velocity are not significant.

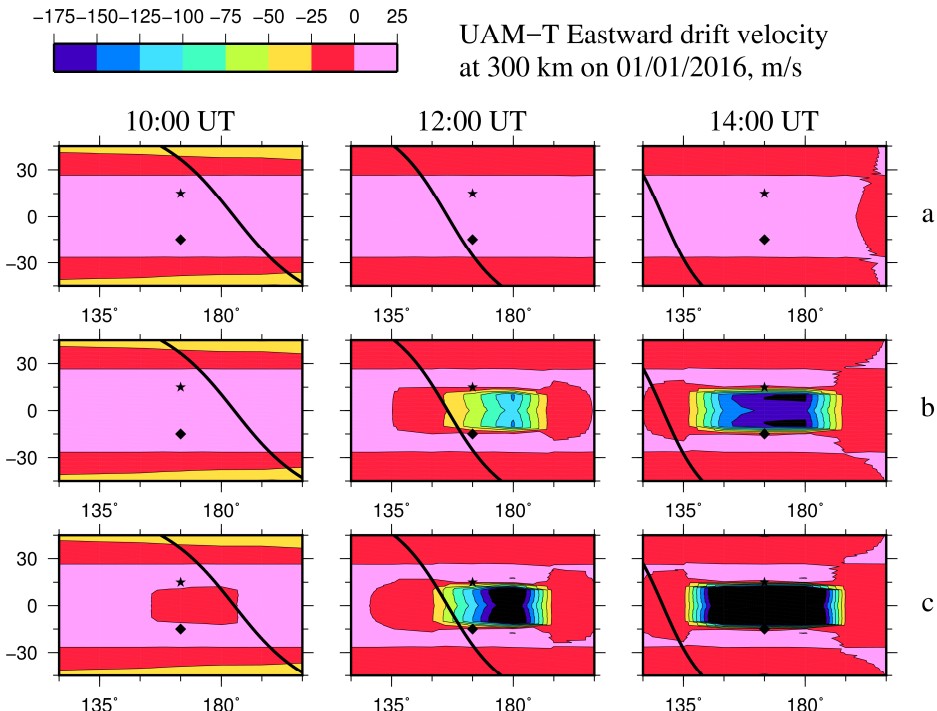

**Figure 3.** The UAM-T calculated eastward drift velocity (m/s) distributions at 300 km in geomagnetic coordinates at 10:00–14:00 UTC on January 1, 2016: (**a**) without seismogenic currents; (**b**) $j_s$ = 20 nA/m$^2$; (**c**) $j_s$ = 40 nA/m$^2$.

The addition of the vertical electric currents leads to a sharp increase in the upward electric field and corresponding zonal drift at the latitudes between the epicenter and conjugated point. For the UAM calculations with the seismogenic vertical electric currents switched on, the resulting westward drift velocity is 3–4 times higher in comparison to the background values calculated without the seismogenic currents. This effect was reproduced by both versions of the UAM. According to our simulation results for this particular case study, an additional zonal drift is more intense than the vertical drift under the action of the zonal electric field, thus, it brings a greater effect to the resulting TEC disturbances.

In the GPS observations, the strong positive TEC disturbances relative to the quiet values are clearly visible in the area northeast of the epicenter (Figure 1a), and, at first glance, they are similar to the ionospheric precursors of earthquakes, but their morphology is not consistent with the previously reported features of pre-seismic TEC disturbances. Firstly, there are no effects near the magnetically conjugated point in the Southern Hemisphere. Secondly, these regions appeared in the GIM-TEC for the first time at 06:00 UT, i.e., earlier that day. Thirdly, they are quite far from the epicenter.

A comparison between the UAM calculation results with the seismic origin electric currents switched on (Figure 1c,d) and off (Figure 1b) shows that these positive GIM-TEC disturbances are not associated with seismogenic currents in accordance with the hypothesis of the aerosols role in the seismo-ionospheric relationship [12–14]. We suppose that these disturbances are related with the high geomagnetic activity. In this period, the ring current heats the outer part of the Earth's plasmasphere. This results in an increase in the downward diffusion plasma flows from the plasmasphere to the ionosphere. These fluxes create the positive GIM-TEC disturbance regions in the Northern Hemisphere due to the larger O/N$_2$ ratio at the ionospheric F2-layer altitudes in winter in comparison with the summer [2].

Thus, the TEC disturbances caused by the upward seismogenic currents present a negative sign in the given conditions of high magnetic activity and low latitudes. Similar regions are clearly visible in the GPS data. The presence of seismogenic currents leads to the increase in the upward electric field and corresponding westward electromagnetic drift of the ionospheric F2-layer plasma. This drift

forms the negative TEC disturbances transporting the plasma from the east to the west through the epicenter meridian, as can be seen in the observations and simulation results.

## 5. Conclusions

This study presents the numerical calculations related to the low latitude ionosphere effects created by the vertical electric currents of seismic origin under high geomagnetic activity. The simulations herein were performed using the 3D global first-principle Upper Atmosphere Model (UAM) and compared with the GIM-TEC data for the high geomagnetic activity period preceding the M6.7 earthquake in India on January 3, 2016.

The UAM calculations reproduced the general behavior of the ionosphere after the main phase of the geomagnetic storm on January 1, 2016 in the form of the negative TEC disturbances propagating from the high latitudes, which were especially strong in the Southern (summer conditions) Hemisphere. It is shown that the seismogenic currents' effects (ionospheric precursors of earthquakes) can be revealed in the background of global geomagnetic disturbances. They are visible as the regions with additional negative TEC disturbances formed on the eastern side of the epicenter meridian and extended to the western side, both in the simulations and observations.

It was found that the vertical electric field, which is the sum of electrostatic and dynamo electric fields, plays a decisive role in the formation of the ionospheric precursors of earthquakes at low latitudes. They are related with the upward electric field and corresponding westward component of the electromagnetic [E × B] drift.

**Author Contributions:** Conceptualization, methodology, A.N.; investigation, M.K. (Mikhail Karpov), M.K. (Maria Knyazeva); visualization, M.K. (Mikhail Karpov), M.K. (Maria Knyazeva); writing—original draft preparation, M.K. (Mikhail Karpov), M.K. (Maria Knyazeva); writing—review and editing, A.N.

**Funding:** This research received no external funding.

**Acknowledgments:** Authors are thankful to the International GNSS Service for providing GPS TEC data (ftp://cddis.gsfc.nasa.gov) and to the World Data Center for Geomagnetism at Kyoto University Japan for providing geomagnetic data (http://swdcwww.kugi.kyoto-u.ac.jp).

**Conflicts of Interest:** The authors declare no conflict of interest.

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
