# Peer review of "Seismogenic Disturbances of the Ionosphere During High Geomagnetic Activity"

_atmosphere, doi:10.3390/atmos10070359_

Round 1

Reviewer 1 Report

Several points need to be clarified before publication of this paper:

As the pre-earthquake period corresponds to a period of high geomagnetic activity, it is very difficult to attribute any change before the EQ to a pre-seismic effect. In particular, when you said “It was shown that the local ionospheric effects of seismic origin can be identified on the background of the global geomagnetic disturbances.” , it is far to be clear.

When you show some data, it only concerns January 1rst but what about the other days before and after the EQ day?

During the discussion about the UAM simulations it is strange that the Equatorial Ionization Anomaly (EIA) is never mentioned. EIA is one of the most important events around the equator. You can see for example “Chen, Y., L. Liu, H. Le, W. Wan, and H. Zhang (2016), Equatorial ionization anomaly in the low-latitude topside ionosphere: Local time evolution and longitudinal difference, J. Geophys. Res. Space Physics, 121, 7166–7182, doi:10.1002/2016JA022394.” But there are a huge number of papers dealing with this event.

Minor points:

Line 44 change

2 equations (4)

Line 121 reference needed

Line 122 carrier

Line 126 zone 1 and zone 2 ???

Author Response

1. Thank you very much for the correct comments. Our major revision consists in the full change of the paper design (color figures) and the method and results description. Our new revised version is based now on the invited review article by Namgaladze [1]. This article is revised according to the requirements of Colin Chen to the end of August 2019.  All your remarks will be taken into account in this invited article. As for identification of the local ionospheric effects of seismic origin on the background of the global geomagnetic disturbances, the key points for dividing them are dimensions, positions and mobility as it is said in lines 37-43: “The main features of the EQ preparation effects in TECs are the locality, connections with the tectonic faults, weak mobility and small occupied areas [1] in comparison with the ionospheric effects of the geomagnetic storms and substorms. The last are usually global, connected with the auroral precipitations and electric fields related to the solar wind, and propagates equatorwards from the auroral zones, sometime in form of TIDs (Travelling Ionosphere Disturbances) due to the AGW (Acoustic Gravity Waves) (Brunelli and Namdaladze, 1988) [2]. Their duration lasts from several hours for substorms to several days for storms”.

2. We see the typical EQ precursor spots in TEC only on January 1rst.

3. We use the words “low latitude ionosphere” instead EIA because the Imphal, India (24.8° N, 93.7° E, mag.lat.15°) EQ location is not purely equatorial.

4. Corrected, including definitions of FAC zone 1 and zone 2 in [1]. The FACs are the Field Aligned Currents, flowing along geomagnetic field lines.  They transfer electric charges from the solar wind to the polar caps boundaries (zone 1) and from the magnetosphere plasma sheet to the auroral zone (zone 2). FACs 2 are opposites to the FAC1.

Reviewer 2 Report

Thr results are interesting and important for a correct individuation of the ionosphere variations possible related to earthquakes.

I hope in more similar results in future!!

Author Response

Thank you very much!

Reviewer 3 Report

This work presents 3D UAM simulation fits of the distribution and time evolution of the TEC over the epicenter of the January 3, 2016 M6.7 EQ in India with only one free parameter, namely the vertical electric current of (seemingly) seismic origin. The agreement with GIM of the TEC is very interesting, the presentation is clear, and the paper deserves publication. I would like the authors to consider the following points:

1) It would be nice to see definitions of various terms (terminator, geomagnetic vs geographical coordinates, geomagnetically conjugate point, parameter Ap, etc.) in the text for the non-experts.

2) Is there an explanation why the UAM-T simulations fail to reproduce the TEC distribution and evolution (this is not mentioned in the text), whereas the UAM-M ones with only one extra parameter do so successfully? Why does the introduction of the Ap parameter make such a big difference? Maybe the authors explained that in the text, but it is not clear to me.

3) It would be nice to have all maps of TEC distributions (both observational and numerical) at the same spatial resolution (2.5o x 5o or so). The comparison would be much easier for the reader.

4) The authors claim that the vertical electric current that they used in their simulations is of seismic origin, and implement a certain distribution along the 4000km long tectonic fault parallel to the 30o meridian. Is that the only possible explanation? How can they account for the geomagnetic activity observed between December 20 and 22, 2015. Is that also due to another EQ in the same (or a different) region?

Author Response

1. These definitions will be done in [1]. It is not necessary to repeat them again in the present paper.

2. This is related with the NRLMSISЕ-00 [8] shortages and the dependence from Ap in NRLMSISE-00.

3. This will be done in [1] via presentation of the global observed and calculated TEC maps examples. The parts of such geomagnetic maps are shown in Figures 1 and 2.

4. We consider that vertical electric current along the tectonic fault parallel to the 30o meridian is the only possible explanation indeed whereas the geomagnetic activity observed between December 20 and 22, 2015 is not related with any EQ.

Round 2

Reviewer 1 Report

Comments on your answers are below.

2. We see the typical EQ precursor spots in TEC only on January 1rst.

This need to be clearly said in the paper, i.e., “we only observe such event on January 1rst at least during X months/days around the EQ day”

3. We use the words “low latitude ionosphere” instead EIA because the Imphal, India (24.8° N, 93.7° E, mag.lat.15°) EQ location is not purely equatorial.

When I check Figure 1 I do not see the EQ epicenter at a longitude of 24° but rather it is around 15° (we do not know if the latitude in Figure 1 is geographic or geomagnetic but at this longitude it is practically the same) !!!

Despite your answer I think you are well concerned by the EIA and its double crest around the magnetic equator. This needs to be clearly mentioned. This does not deserved your study as it has been already shown that EQ can influence the EIA (see for example some papers from Pulinets).

By the way it will be also interesting to give the relation between UT and LT in Figure 1.

Author Response

Our first reply. 2. We see the typical EQ precursor spots in TEC only on January 1rst.

Reviewer: This need to be clearly said in the paper, i.e., “we only observe such event on January 1rst at least during X months/days around the EQ day”

New reply: We have added the recommended remark to our paper (lines 109-111) of the last paper version).

Our first reply. 3. We use the words “low latitude ionosphere” instead EIA because the Imphal, India (24.8° N, 93.7° E, mag.lat.15°) EQ location is not purely equatorial.

Reviewer: When I check Figure 1 I do not see the EQ epicenter at a longitude of 24° but rather it is around 15° (we do not know if the latitude in Figure 1 is geographic or geomagnetic but at this longitude it is practically the same) !!!

Despite your answer I think you are well concerned by the EIA and its double crest around the magnetic equator. This needs to be clearly mentioned. This does not deserved your study as it has been already shown that EQ can influence the EIA (see for example some papers from Pulinets).

By the way it will be also interesting to give the relation between UT and LT in Figure 1.

New reply: On the capture to the Figure 1 it is written that the coordinates on the Figures 1 and 2 are geomagnetic. The coordinates of the Imphal, India (24.8° N, 93.7° E) are geographical. Letter N denotes north geographical latitude; letter E denotes east geographical longitude. The relation between UT and LT in Figure 1 can be seen using the terminator (black line). There is the sunlit area on the left of it and the night side on the right of the terminator.

Of course, EQs influenced on EIA and we write on this matter in all our papers. The Imphal is located near the north crest of the EIA, and we added these remarks to our paper (lines 60-62 and 171-172 of the last paper version).